# Effect of linezolid on platelet count in critically ill patients with thrombocytopenia

**Hiroomi Tatsumi** [ID]*[º]**, Masayuki Akatsuka**[º]**, Hiromitsu Kuroda**[ID]**, Satoshi Kazuma, Shintaro Suzuki, Yoshiki Masuda**

Department of Intensive Care Medicine, Sapporo Medical University School of Medicine, Sapporo, Hokkaido, Japan

º These authors contributed equally to this work.

\* htatsumi@sapmed.ac.jp

## Abstract

### Introduction

Linezolid (LZD) is one of the antibiotics used to treat methicillin-resistant *Staphylococcus aureus*. In Japan, the dose of LZD is not generally adjusted by renal function or therapeutic drug monitoring and is readily available for critically ill patients. The adverse effects of LZD include pancytopenia, especially thrombocytopenia. We investigated the effect of LZD on platelet counts in critically ill patients with thrombocytopenia during admission to the intensive care unit (ICU).

### Methods

Fifty-five critically ill patients with existing thrombocytopenia (platelet count < 100 ×10³ /µL) who received LZD for five days or more during the period from January 2011 to October 2018 were included. Changes in platelet count and frequency of platelet concentrate (PC) transfusion were evaluated retrospectively.

### Results

Mean (± standard error) platelet count prior to initiation of LZD was 47 ± 4 ×10³ /uL, which increased significantly to 86 ± 13 ×10³ /uL on day 15 (p<0.01). Median [interquartile range] duration of LZD therapy was 9 [8–12] days. Thirty-two patients (58.2%) required PC transfusion in the 15-day study period. The daily rate of PC transfusion decreased from 30.2% on days 1–5 to 18.2% on days 11–15. Similar tendencies were observed in patients with non-hematological and hematological disease.

### Conclusion

Thrombocytopenia in critically ill patients in the ICU did not worsen after initiation of LZD therapy, and may be considered for the treatment of MRSA in this setting.

**Data Availability Statement:** All relevant data are within the paper and its Supporting Information files.

**Funding:** The authors received no specific funding for this work.

**Competing interests:** Tatsumi H received lecture fees from TSUMURA & CO and industry-academia collaborative research grant from Otsuka Pharmaceutical Factory, Inc. Masuda Y received lecture fees from MSD K.K., Asahi Kasei Pharma Corp. and Japan Blood Product and industry-academia collaborative research grant from JIMRO Co., Ltd. Other authors declare that they have no competing interests. This does not alter our adherence to PLOS ONE policies on sharing data and materials.

**Abbreviations:** MRSA, methicillin-resistant *Staphylococcus aureus*; VRE, vancomycin-resistant enterococci; LZD, linezolid; VAN, vancomycin; TDM, therapeutic drug monitoring; AKI, acute kidney injury; CRRT, continuous renal replacement therapy; DIC, disseminated intravascular coagulation; ICU, intensive care unit; PC, platelet concentrate; IQR, interquartile range; APACHE, acute physiology and chronic health evaluation; SOFA, sequential organ failure assessment; SE, standard error.

# Introduction

Among Gram-positive bacteria, methicillin-resistant *Staphylococcus aureus* (MRSA) and vancomycin-resistant *enterococci* (VRE) are crucial pathogens that cause hospital-acquired infection [1] and are related to high rates of hospital mortality, especially in critically ill patients [2]. Of the antibiotics used for MRSA and VRE, linezolid (LZD) has several features including low molecular weight (337.35 Da), low protein binding rate (31%), large distribution volume (40–50 L/kg), and lipophilicity, and therefore excellent tissue migration properties. Among the various anti-MRSA drugs, which include vancomycin (VAN), in Japan the dose of LZD is not generally adjusted based on renal function or therapeutic drug monitoring (TDM). Therefore, LZD is readily available for patients with acute kidney injury (AKI) and those who are receiving continuous renal replacement therapy (CRRT). One of the adverse effects of LZD is pancytopenia, especially thrombocytopenia. Previous studies have shown that the development of LZD-induced thrombocytopenia may be associated with body weight [3–5], baseline platelet level [4, 6–8], prolonged duration of therapy [9, 10], renal function [3, 5, 6, 11, 12] and plasma LZD concentration [4, 8, 11]. Accordingly, an anti-MRSA drug other than LZD is commonly selected for patients with thrombocytopenia, without regard to other causes of thrombocytopenia such as sepsis, disseminated intravascular coagulation (DIC), and myelosuppression. The aim of this study was to investigate the effect of LZD on platelet count in critically ill patients with thrombocytopenia in the intensive care unit (ICU) setting.

# Patients and methods

## Study design

This retrospective study was performed at the ICU of Sapporo Medical University Hospital (Sapporo, Japan). The study protocol conformed to the ethical guidelines enshrined in the Declaration of Helsinki and was approved by the Research Ethics Committee of Sapporo Medical University (approval No. 302–145). The requirement for informed consent was waived due to the retrospective and observational nature of the study. An information disclosure document about the study was created and made available to the study patients and their families via the website of the hospital's ICU, guaranteeing the opportunity for study patients and their families to refuse participation.

## Patients and setting

The subjects were critically ill patients in the ICU with thrombocytopenia who received LZD for five days or more during the period from January 2011 to October 2018. Patients with sepsis who are admitted to the ICU often have AKI or undergo CRRT, and when selecting an anti-MRSA drug that requires dose adjustment, it takes time to adjust the blood concentration within the effective range. Because the dose of LZD is not generally adjusted based on renal function or TDM in Japan, we chose LZD for the treatment of MRSA. In this study, thrombocytopenia was defined as a platelet count of less than $100 \times 10^3$ /μL at the time of LZD administration. Patients who died due to acute or irreversible progression of the primary disease (Fig 1) within 15 days after the start of LZD administration and those younger than 18 years were excluded. As a general rule, the daily dose of LZD was 600 mg intravenously twice a day, and the timing of LZD discontinuation was decided by physicians and ICU staff at the daily ICU conference. The use of platelet concentrate (PC) transfusion was also decided at the daily ICU conference, with indications including the platelet count, change in the platelet count, presence or absence of bleeding, treatment content such as blood purification therapy, and the necessity of treatment with a high bleeding risk.

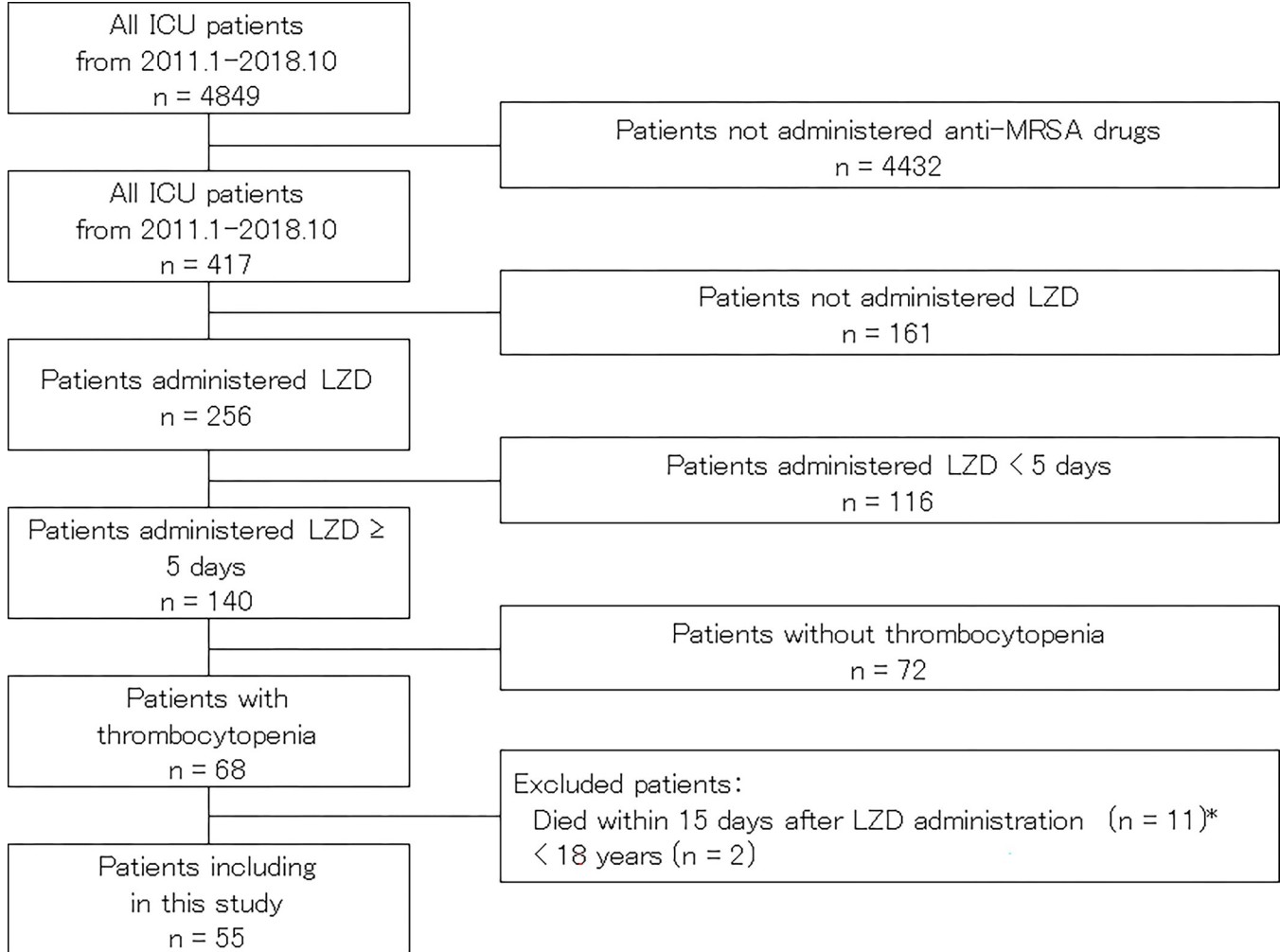

**Fig 1. Flow chart of patient selection.** *Cause of death: Complications after hematopoietic stem cell transplantation: 4 Fulminant hepatitis / acute liver failure: 3 Postresuscitation encephalopathy: 2 Alveolar hemorrhage due to vasculitis: 1 Progression of malignant tumor: 1.

Patient characteristics including age and sex, primary disease, reason for ICU admission, change in platelet count from days 1 to 15 after the start of LZD administration, and PC transfusion from days 1 to 15 after the start of LZD administration were evaluated retrospectively. The daily rate of PC transfusion (defined as PC transfused patients divided by the total patients) was calculated over the 15 days and compared among three periods: early period (days 1–5), middle period (days 6–10), and late period (days 11–15).

Because the present study was performed retrospectively, daily platelet counts had not been obtained in all patients, and the original data had many missing values. The missing data were mostly after ICU discharge, especially on the first day after PC transfusion, when the platelet count was expected to increase. Therefore, we prepared and evaluated corrected data in which the missing platelet count data were compensated for as the average of the measured values on the previous day and the next day.

The patients were divided into two groups: a hematological disease group and a non-hematological disease group (Table 1). In the hematological disease group, thrombocytopenia or pancytopenia occurred due to chemotherapy or hematopoietic stem cell transplantation for

**Table 1. Patient characteristics (January 2011 to October 2018, Sapporo Medical University Hospital).**

| | | | All | | Non-hematological | | Hematological | |
|---|---|---|---|---|---|---|---|---|
| | | | (n = 55) | | (n = 41) | | (n = 14) | |
| Age | (y): median [IQR] | | 69 | [62.5–75.5] | 73 | [66–76] | 50 | [61.8–75.5] |
| Sex | | | | | | | | |
| | Female: N (%) | | 16 | (29.1) | 12 | (29.3) | 4 | (28.6) |
| | Male: N (%) | | 39 | (70.9) | 29 | (70.7) | 10 | (71.4) |
| Primary disease | | | | | | | | |
| | Non-hematological: N (%) | | 41 | (74.5) | | | | |
| | | Cardiovascular: N (%) | | | 15 | (27.3) | | |
| | | Gastrointestinal: N (%) | | | 9 | (16.4) | | |
| | | Hepatobiliary-pancreatic: N (%) | | | 6 | (10.9) | | |
| | | Respiratory: N (%) | | | 3 | (5.5) | | |
| | | Head and neck: N (%) | | | 2 | (3.6) | | |
| | | Orthopedic: N (%) | | | 2 | (3.6) | | |
| | | Other: N (%) | | | 4 | (7.3) | | |
| | Hematological: N (%) | | 14 | (25.5) | | | | |
| | | Acute myeloid leukemia: N (%) | | | | | 6 | (10.9) |
| | | Acute lymphoblastic leukemia: N (%) | | | | | 4 | (7.3) |
| | | Myelodysplastic syndromes: N (%) | | | | | 2 | (3.6) |
| | | Other: N (%) | | | | | 2 | (3.6) |
| APACHE II: median [IQR] | | | 24 | [20.5–28] | 25 | [19–28] | 23.5 | [21–29.5] |
| SOFA score at ICU admission: median [IQR] | | | 8 | [7–10.5] | 8 | [6–10] | 9 | [8–10.8] |
| Presumed cause of thrombocytopenia | | | | | | | | |
| | Sepsis: N (%) | | 26 | (47.3) | 26 | (63.4) | 0 | |
| | Chemotherapy: N (%) | | 2 | (3.6) | 1 | (2.4) | 1 | (7.1) |
| | Chemotherapy + sepsis: N (%) | | 11 | (20.0) | 7 | (17.1) | 4 | (28.6) |
| | Hepatic failure: N (%) | | 1 | (1.8) | 1 | (2.4) | 0 | |
| | Hepatic failure + sepsis: N (%) | | 4 | (7.3) | 4 | (9.8) | 0 | |
| | Hematopoietic stem cell transplantation: N (%) | | 4 | (7.3) | 0 | | 4 | (28.6) |
| | Hematopoietic stem cell transplantation + sepsis: N (%) | | 4 | (7.3) | 0 | | 4 | (28.6) |
| | Other: N (%) | | 3 | (5.5) | 2 | (4.9) | 1 | (7.1) |
| Infection on ICU admission: N (%) | | | 34 | (61.8) | 25 | (61.0) | 9 | (64.3) |
| DIC on ICU admission: N (%) | | | 26 | (47.3) | 19 | (46.3) | 7 | (50.0) |
| AKI on ICU admission: N (%) | | | 31 | (56.4) | 23 | (56.1) | 8 | (57.1) |
| Duration of LZD administration (day): median [IQR] | | | 9 | [8–12] | 9 | [8–12] | 10 | [7.3–12.8] |
| Administration of CRRT: N (%) | | | 42 | (76.4) | 32 | (78.0) | 10 | (71.4) |
| Administration of PC transfusion: N (%) | | | 32 | (58.2) | 19 | (46.3) | 13 | (92.9) |
| Duration of ICU stay (d): median [IQR] | | | 15 | [8.5–23.5] | 15 | [8–22] | 16.5 | [9.3–23.8] |
| ICU mortality: N (%) | | | 13 | (23.6) | 8 | (19.5) | 5 | (35.7) |
| 28-day mortality: N (%) | | | 16 | (29.1) | 9 | (22.0) | 7 | (50.0) |

the treatment of hematological diseases or malignancies. Change in platelet count during LZD administration was analyzed in each group.

## Statistical analysis

Categorical variables and continuous variables of the patients' characteristics are presented as the number (%) and median [interquartile range, IQR], respectively. Platelet counts are

presented as the mean ± standard error. Serial change in platelet count was assessed using Friedman's test. When a statistically significant difference was obtained after Friedman's test, Wilcoxon's t-test with Bonferroni correction was performed as a post hoc test. Probability values less than 0.05 in a two-sided test were considered statistically significant.

## Results

### Patient characteristics

Patient characteristics are listed in Table 1. Fifty-five patients were included (Fig 1). LZD was administered as empiric or definitive therapy. Median age was 69 [62.5–75.5] years and 39/55 (70.9%) were male. As indicators of the severity of illness, acute physiology and chronic health evaluation (APACHE) II score and sequential organ failure assessment (SOFA) score were 24 [20.5–28] and 8 [7–10.5], respectively. Infection was diagnosed on admission to the ICU in 34/55 (61.8%), and 26/55 (47.3%) developed DIC on ICU admission. The effects of chemotherapy and hematopoietic stem cell transplantation were also taken into consideration as causes of thrombocytopenia, in addition to the effects of sepsis alone. Among the primary diseases, cardiovascular disease was the most common (27.3%), followed by hematological (25.4%) and gastrointestinal disease (16.4%). Thirty-one patients (56.4%) had AKI on ICU admission and 42 patients (76.4%) needed CRRT while in the ICU. Median duration of LZD administration was 9 [8–12] days and median duration of ICU stay was 15 [8.5–23.5] days.

### Changes in platelet count and PC transfusion in all patients

Changes in mean platelet count are shown for the original and corrected data in Fig 2A and 2B, respectively. The mean (± standard error, SE) platelet count was $47 \pm 4 \times 10^3$ /µL before LZD administration, which showed a gradual but statistically significant increase over the 15-day period (p < 0.01, in the original and corrected data). Mean (± SE) platelet counts were $86 \pm 13 \times 10^3$ /µL in the original data and $95 \pm 13 \times 10^3$ /µL in the corrected data on day 15, which were significantly higher than those before initiation of LZD therapy (p < 0.01 in both

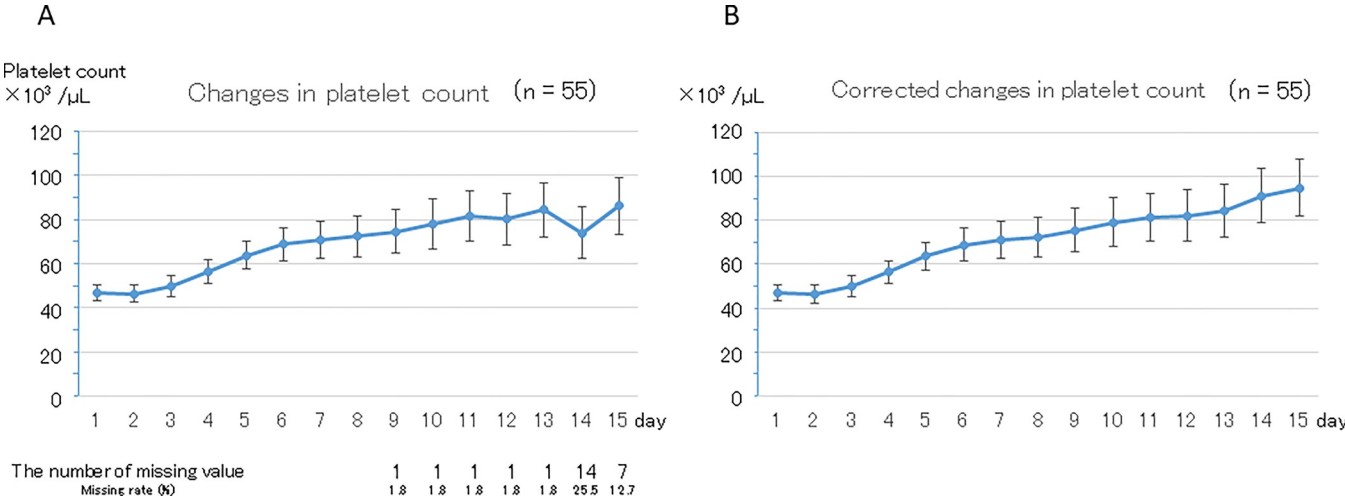

**Fig 2. Change in platelet count in all patients (January 2011 to October 2018, Sapporo Medical University Hospital).** A. Original data. Platelet count showed a gradual but statistically significant increase (p < 0.01). Platelet count was significantly higher on day 15 than before LZD administration ($86 \pm 13 \times 10^3$ /µL and $47 \pm 4 \times 10^3$ /µL, respectively; p < 0.01). There were missing values after the 9th day. B. Corrected data. Platelet count showed a gradual but statistically significant increase (p < 0.01). Platelet counts were significantly higher on day 15 than before LZD administration ($95 \pm 13 \times 10^3$ /µL and $47 \pm 4 \times 10^3$ /µL, respectively; p < 0.01).

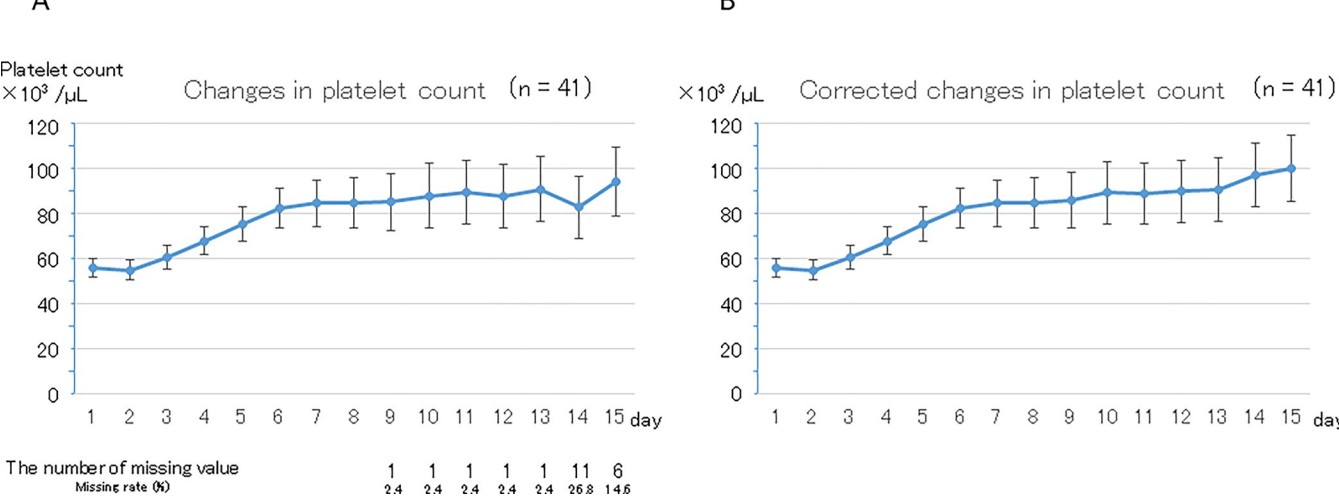

**Fig 3. Change in platelet count in the non-hematological disease group (January 2011 to October 2018, Sapporo Medical University Hospital).** A. Original data. Platelet count showed a gradual but statistically significant increase (p < 0.01). Platelet count was significantly higher on day 15 than before LZD administration (94 ± 16 ×10³ /μL and 56 ± 4 ×10³ /μL, respectively; p = 0.02). There were missing values after the 9th day. B. Corrected data. Platelet count showed a gradual but statistically significant increase (p < 0.01). Platelet count was significantly higher on day 15 than before LZD administration (100 ± 15 ×10³ /μL and 56 ± 4 ×10³ /μL, respectively; p < 0.01).

the original and corrected data). Thirty-two patients (58.2%) required PC transfusion in the 15 days after starting LZD administration. The daily rates of PC transfusion after starting LZD administration were 30.2%, 20.7%, and 18.2% in the early, middle, and late periods, respectively.

## Changes in platelet count and PC transfusion in the non-hematological disease group

Sub-analyses of the 41 patients whose primary disease was non-hematological are shown in Fig 3. Findings in this group were similar to those in all patients even when hematological diseases were excluded. Mean (± SE) platelet count was 56 ± 4 ×10³ /μL before LZD administration, and increased gradually over 15 days (p < 0.01 in both the original and corrected data). Mean (± SE) platelet counts were 94 ± 16 ×10³ /μL in the original data and 100 ± 15 ×10³ /μL in the corrected data on day 15, which were significantly higher than those before initiation of LZD therapy (p = 0.02 and p < 0.01, respectively). Nineteen patients (46.3%) required PC transfusion in the 15 days after starting LZD administration. The daily rates of PC transfusion after starting LZD administration were 17.1%, 8.8%, and 7.3% in the early, middle, and late periods, respectively.

## Changes in platelet count and PC transfusion in the hematological disease group

Sub-analyses of the 14 patients whose primary disease was hematological are shown in Fig 4. A similar tendency was observed even when the primary disease was limited to hematological disease. Mean (± SE) platelet count was 20 ± 2 ×10³ /μL before LZD administration and increased gradually over 15 days (p < 0.01 in both the original and corrected data). Mean (± SE) platelet counts were 66 ± 24 ×10³ /μL in the original data and 80 ± 26 ×10³ /μL in the corrected data on day 15, the latter being significantly higher than that before initiation of LZD therapy (p = 0.04). Thirteen patients (92.9%) required PC transfusion in the 15 days after

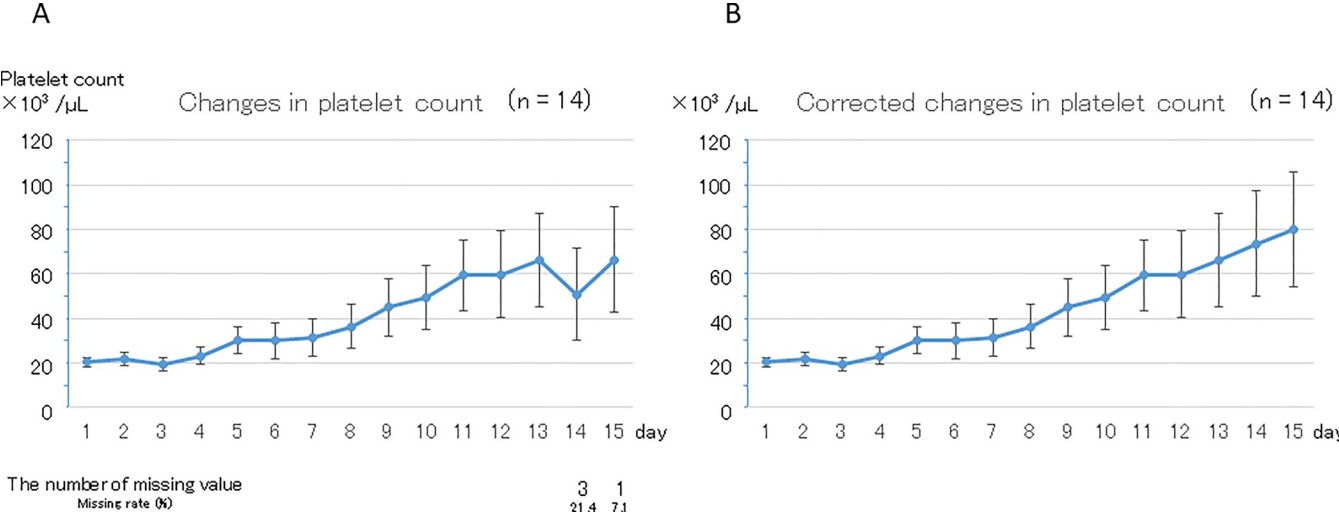

**Fig 4. Change in platelet count in the hematological disease group (January 2011 to October 2018, Sapporo Medical University Hospital).** A. Original data. Platelet count showed a gradual but statistically significant increase (p < 0.01). Platelet count was significantly higher on day 15 than before LZD administration (66 ± 24 ×10$^3$ /μL and 20 ± 2 ×10$^3$ /μL, respectively; p = 0.08). There were missing values after the 14th day. B. Corrected data. Platelet count showed a gradual but statistically significant increase (p < 0.01). Platelet count was significantly higher on day 15 than before LZD administration (80 ± 26 ×10$^3$ /μL and 20 ± 2 ×10$^3$ /μL, respectively; p = 0.04).

starting LZD administration. The daily rates of PC transfusion after starting LZD administration were 68.6%, 55.7%, and 50.0% in the early, middle, and late periods, respectively.

## Changes in platelet count in patients without PC transfusion

Sub-analysis in 23 patients who did not receive PC transfusion are shown in Fig 5. Mean (± SE) platelet count was 68 ± 5 ×10$^3$ /μL before LZD administration, and increased gradually

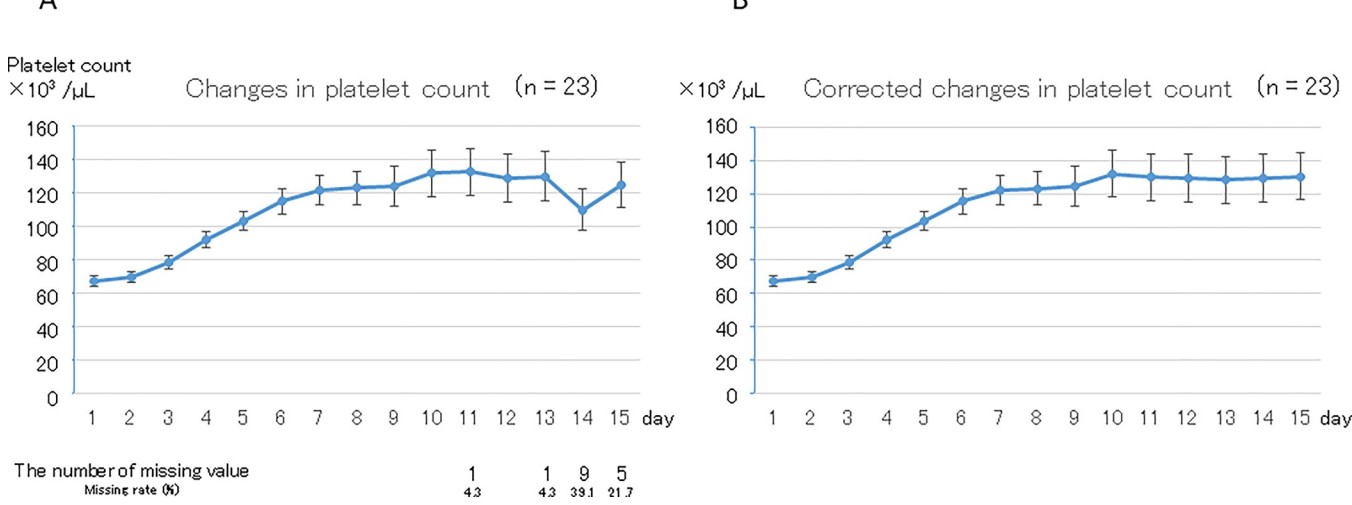

**Fig 5. Change in platelet count in patients without PC transfusion (January 2011 to October 2018, Sapporo Medical University Hospital).** A. Original data. Platelet count showed a gradual but statistically significant increase (p < 0.01). Platelet count was significantly higher on day 15 than before LZD administration (125 ± 21×10$^3$ /μL and 68 ± 5 ×10$^3$ /μL, respectively; p = 0.03). There were missing values after the 9th day. B. Corrected data. Platelet count showed a gradual but statistically significant increase (p < 0.01). Platelet count was significantly higher on day 15 than before LZD administration (130 ± 22 ×10$^3$ /μL and 68 ± 5 ×10$^3$ /μL, respectively; p = 0.04).

over 7 days and then plateaued (p < 0.01 in both the original and corrected data). Mean (± SE) platelet counts were $125 \pm 21 \times 10^3$ /μL in the original data and $130 \pm 22 \times 10^3$ /μL in the corrected data on day 15, which were significantly higher than those before initiation of LZD administration (p = 0.03 and 0.04, respectively).

## Discussion

Numerous reports [9, 10, 13, 14] have shown that long-term administration of LZD is associated with thrombocytopenia. In general, LZD-induced thrombocytopenia occurs from 7 days after the start of administration and resolves several days after discontinuation of LZD [2, 6, 13–18]. However, the effect of administration of LZD on platelet count in patients with thrombocytopenia was previously unknown. The present study showed that platelet count increased gradually and that the need for PC transfusion showed a decreasing trend after LZD administration in patients whose platelet count was less than $100 \times 10^3$ /μL in the early period of LZD administration. These findings were similar between the hematological and non-hematological disease groups. In the subanalysis of patients without PC transfusion, platelet count increased gradually after LZD administration and plateaued after day 7. Therefore, we consider that LZD does not cause thrombocytopenia within at least the first 10 days of administration, and may be considered for patients with thrombocytopenia due to infection or hematological disorder. In addition, we observed no leukopenia or erythrocytopenia that could be attributed to LZD in the present patients.

We consider that in the present study, the platelet count increased as a result of effective treatment (including LZD administration) in critically ill patients with thrombocytopenia. Administration of LZD may be associated with resolution of infection and recovery from serious illness, resulting in an increase in platelet count. Although thrombocytopenia may occur as a side effect of LZD in some patients, in a greater number of patients with serious illness, improvements due to the efficacy of LZD occurred earlier than the LZD-induced thrombocytopenia. In fact, the median duration of LZD was 9 days in the present study, which was relatively short. We consider that LZD-induced myelosuppression occurred in few patients because the platelet count recovered before LZD-induced thrombocytopenia would be expected to occur, even after administration of LZD. In other words, if early administration of LZD is effective and LZD is discontinued within a short term, thrombocytopenia due to bone marrow suppression caused by LZD may be avoidable. Even in critically ill patients with thrombocytopenia, initiation of LZD may be considered when LZD is necessary to treat serious illness due to MRSA infection.

It is well known that AKI is more likely to occur in critically ill patients. As anti-MRSA drugs are generally excreted by the kidneys, the dose of VAN or teicoplanin (which requires TDM) depends on the degree of impairment of renal function. When AKI worsens to the point that CRRT is required, the dosage decisions become more complicated. LZD has a wide safety margin in terms of blood concentration, and LZD dose is not generally adjusted by renal function or TDM in Japan, even in patients with AKI or during CRRT. Therefore, LZD is easy to use as anti-MRSA drug in treatment of critically ill patients. However, renal failure has been reported as a key determinant of whether thrombocytopenia occurs after LZD administration [19]. In fact, more than half of the present patients had AKI on ICU admission, and 76% required CRRT while in the ICU. Moreover, it has been reported that there is an increasing need for dose adjustment due to individual differences in the efficacy and toxicity of LZD [20]. The effect of renal function and CRRT on LZD concentration was not clear in the present patients. However, if TDM becomes common in Japan in the future, the safety and efficacy of LZD could possibly be improved by adjusting the dose of LZD.

Despite the rapid effectiveness of LZD due to its pharmacokinetic features, there may be a hesitancy to administer LZD in patients with thrombocytopenia, especially in those with hematological disease. However, the present finding that platelet count was not affected by LZD during a 2-week administration period suggests that LZD could be selected for empiric therapy even in patients with thrombocytopenia. In particular, if LZD administration is discontinued shortly after the causative bacteria is confirmed, it appears that LZD can be administered safely regardless of thrombocytopenia. In the future, it will be necessary to conduct a prospective and multicenter study to clarify the safety of LZD in terms of platelet count in patients with thrombocytopenia.

The present study has several limitations. First, this study was a retrospective, single arm, observational study conducted at a single institute. We chose a single arm study design because there were insufficient patients receiving anti-MRSA drugs other than LZD with which to compare data. Second, platelet counts were missing in several of the data sets, which was compensated for by calculating the average of measured values from the previous and subsequent days. Third, the cause of thrombocytopenia before LZD administration was related to several factors, such as infection and administration drugs, and could not be accurately classified. Fourth, it is undeniable that PC transfusion would have resulted in a sustained increase in platelet counts in some patients; however, as the frequency of PC transfusion progressively decreased from the early to late periods, we consider that the platelet count was recovering even during administration of LZD. In addition, LZD treated the underlying infection as the cause of the initial thrombocytopenia in a proportion of the patients. Therefore, in certain patients, both the PC transfusions and the LZD therapy may have led to a gradual increase in platelet count.

## Conclusion

Initiation of LZD therapy did not worsen thrombocytopenia in critically ill patients in the ICU, and may therefore be considered for the treatment of MRSA in this setting. We recommend close monitoring of platelet count and prompt discontinuation of LZD after resolution of the infection.

## Supporting information

**S1 File.**
(XLSX)

## Author Contributions

**Conceptualization:** Hiroomi Tatsumi.

**Data curation:** Hiroomi Tatsumi, Masayuki Akatsuka, Hiromitsu Kuroda, Satoshi Kazuma, Shintaro Suzuki.

**Investigation:** Hiroomi Tatsumi.

**Methodology:** Hiroomi Tatsumi.

**Project administration:** Hiroomi Tatsumi.

**Supervision:** Yoshiki Masuda.

**Writing – original draft:** Hiroomi Tatsumi.

**Writing – review & editing:** Masayuki Akatsuka, Hiromitsu Kuroda, Satoshi Kazuma, Yoshiki Masuda.

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
