## [Decision Letter · Decision Letter 0]

29 Oct 2022

PONE-D-22-25412Effect of linezolid on platelet count in critically ill patients with thrombocytopeniaPLOS ONE

Dear Dr. Tatsumi,

Thank you for submitting your manuscript to PLOS ONE. After careful consideration, we feel that it has merit but does not fully meet PLOS ONE’s publication criteria as it currently stands. Therefore, we invite you to submit a revised version of the manuscript that addresses the points raised during the review process.

Both reviewers felt that this manuscript was of interest in patients on linezolid. Both reviewers did highlight a number of minor errors. In  addition, there were 2 queries raised by reviewers one - specifically there appears to be a discrepancy between the results presented in the abstract and the results presented in the text and this should be corrected. In addition, the reviewer felt that the exclusion of patients who died before the 15 day window might mask some toxicity. It is advised that these results are analysed, possibly as a subgroup

We look forward to receiving your revised manuscript.

Kind regards,

Elizabeth S. Mayne, M.D.

Academic Editor

PLOS ONE

Journal Requirements:

"I have read the journal's policy and the authors of this manuscript have the following competing interests: H.T. received lecture fees from TSUMURA & CO. and industry-academia collaborative research grant from Otsuka Pharmaceutical Factory, Inc. Y.M. received lecture fees from MSD K.K., Asahi Kasei Pharma Corp. and Japan Blood Product and industry-academia collaborative research grant from JIMRO Co., Ltd. Other authors declare that they have no competing interests."

Additional Editor Comments:

Both reviewers felt that this manuscript was of interest in patients on linezolid. Both reviewers did highlight a number of minor errors. In addition, there were 2 queries raised by reviewers one - specifically there appears to be a discrepancy between the results presented in the abstract and the results presented in the text and this should be corrected. In addition, the reviewer felt that the exclusion of patients who died before the 15 day window might mask some toxicity. It is advised that these results are analysed, possibly as a subgroup

Reviewers' comments:

Reviewer's Responses to Questions

**Comments to the Author**

1. Is the manuscript technically sound, and do the data support the conclusions?

Reviewer #1: Yes

Reviewer #2: Yes

2. Has the statistical analysis been performed appropriately and rigorously? 

Reviewer #1: Yes

Reviewer #2: Yes

3. Have the authors made all data underlying the findings in their manuscript fully available?

Reviewer #1: No

Reviewer #2: Yes

4. Is the manuscript presented in an intelligible fashion and written in standard English?

Reviewer #1: Yes

Reviewer #2: Yes

5. Review Comments to the Author

Reviewer #1: This study assessed serial platelet counts in patients treated with Linezolid who had baseline thrombocytopenia. The study is of interest, as Linezolid may cause thrombocytopenia, which can cause hesitancy among prescribers to use this agent in patients with low platelet counts. The study shows evidence that the use of Linezolid is safe in this setting, but it must be emphasized that this was with supportive platelet transfusions in at least half the patients. It is unfortunate that it was not possible to compare this cohort to a group of patients with MRSA infection and thrombocytopenia who were not treated with Linezolid, as the current findings leave the question of Linezolid’s contribution to the need for platelet transfusion unanswered. In addition, there are a few areas which could be improved as detailed below:

1) In the abstract, it is stated that 58% of patients required PC, but in the Table, it is reported as 49%. Please correct the error wherever it lies.

2) Patients who died within 15 days of Linezolid administration were excluded, which may mask serious deleterious effects of Linezolid therapy. Ideally, these patients should be included in a subanalysis, and their causes of death assessed (were there any death due to serious bleeding for example, and what were their platelet counts on demise).

3) It would be interesting to include a subanalysis on the platelet trajectory in patients who did not receive platelet transfusion. An improvement in the platelet count independently from transfusion would further support the safety of Linezolid in these patients.

4) Parts of the manuscript (particularly the discussion) require English language editing, and there are several minor typographical errors. Please make the tracked changes in the attached Word document.

Reviewer #2: A valuable study to begin the discussion on use of LZD in critically ill patients. There are multiple small editorial changes that I have recommended. The discussion needs to be reworked to assist with clarity and readability of points made.

6. PLOS authors have the option to publish the peer review history of their article (what does this mean?). If published, this will include your full peer review and any attached files.

Reviewer #1: No

Reviewer #2: No

---

## [Author Response · Author response to Decision Letter 0]

26 Jan 2023

Author response to Reviewer #1:

Thank you for your detailed and careful review of our original article. We respond to review comments.

1) In the abstract, it is stated that 58% of patients required PC, but in the Table, it is reported as 49%. Please correct the error wherever it lies.

 The numbers in the Table 1 were wrong. Changed All to 32 in Table 1 and Non-hematological to 19 in Result and Table 1.

2) Patients who died within 15 days of Linezolid administration were excluded, which may mask serious deleterious effects of Linezolid therapy. Ideally, these patients should be included in a subanalysis, and their causes of death assessed (were there any death due to serious bleeding for example, and what were their platelet counts on demise).

 All death cases within 15 days after the start of LZD were due to acute or irreversible progression of the primary disease. The duration of LZD administration was 7 [6-8] days. Many patients had low platelet counts at the time of death, but because the median duration of LZD administration was 7 days, we determined that the progression of the primary disease had a significant impact. 

Inserted the following sentence: 

Patients who died due to acute or irreversible progression of the primary disease within 15 days…

And, added the cause of death to Figure legends of Fig. 1.

3) It would be interesting to include a subanalysis on the platelet trajectory in patients who did not receive platelet transfusion. An improvement in the platelet count independently from transfusion would further support the safety of Linezolid in these patients.

 Sub-analysis in patients who did not receive PC transfusion was added in Result, Fig 5A/5B and Discussion. Similar tendency was observed. 

4) Parts of the manuscript (particularly the discussion) require English language editing, and there are several minor typographical errors. Please make the tracked changes in the attached Word document.

 We confirmed and revised as much as possible.

---

## [Decision Letter · Decision Letter 1]

23 Feb 2023

PONE-D-22-25412R1Effect of linezolid on platelet count in critically ill patients with thrombocytopeniaPLOS ONE

Dear Dr. Tatsumi,

Thank you for submitting your manuscript to PLOS ONE. After careful consideration, we feel that it has merit but does not fully meet PLOS ONE’s publication criteria as it currently stands. Therefore, we invite you to submit a revised version of the manuscript that addresses the points raised during the review process.

The reviewer has commented that they made significant edits to improve the quality of the written English but that these were not accepted by the authors. Please can they ensure that all corrections are made prior to resubmission as the quality of the English remains poor.

We look forward to receiving your revised manuscript.

Kind regards,

Elizabeth S. Mayne, M.D.

Academic Editor

PLOS ONE

Journal Requirements:

Additional Editor Comments:

The reviewer has commented that they made significant edits to improve the quality of the written English but that these were not accepted by the authors. Please can they ensure that all corrections are made prior to resubmission as the quality of the English remains poor.

Reviewers' comments:

Reviewer's Responses to Questions

**Comments to the Author**

1. If the authors have adequately addressed your comments raised in a previous round of review and you feel that this manuscript is now acceptable for publication, you may indicate that here to bypass the “Comments to the Author” section, enter your conflict of interest statement in the “Confidential to Editor” section, and submit your "Accept" recommendation.

Reviewer #1: All comments have been addressed

2. Is the manuscript technically sound, and do the data support the conclusions?

Reviewer #1: Yes

3. Has the statistical analysis been performed appropriately and rigorously? 

Reviewer #1: I Don't Know

4. Have the authors made all data underlying the findings in their manuscript fully available?

Reviewer #1: No

5. Is the manuscript presented in an intelligible fashion and written in standard English?

Reviewer #1: No

6. Review Comments to the Author

Reviewer #1: The authors have made improvements to the manuscript, and have largely addressed my major concerns. However, the manuscript (particularly the Discussion) requires English language editing to be fit for publication.

7. PLOS authors have the option to publish the peer review history of their article (what does this mean?). If published, this will include your full peer review and any attached files.

Reviewer #1: No

---

## [Author Response · Author response to Decision Letter 1]

25 Mar 2023

I revised the text and had it checked again by a native English speaker.

---

## [Decision Letter · Decision Letter 2]

9 May 2023

Effect of linezolid on platelet count in critically ill patients with thrombocytopenia

PONE-D-22-25412R2

Dear Dr. Tatsumi,

We’re pleased to inform you that your manuscript has been judged scientifically suitable for publication and will be formally accepted for publication once it meets all outstanding technical requirements.

Kind regards,

Elizabeth S. Mayne, M.D.

Academic Editor

PLOS ONE

Additional Editor Comments (optional):

Reviewers' comments:

Reviewer's Responses to Questions

**Comments to the Author**

1. If the authors have adequately addressed your comments raised in a previous round of review and you feel that this manuscript is now acceptable for publication, you may indicate that here to bypass the “Comments to the Author” section, enter your conflict of interest statement in the “Confidential to Editor” section, and submit your "Accept" recommendation.

Reviewer #1: All comments have been addressed

Reviewer #3: All comments have been addressed

2. Is the manuscript technically sound, and do the data support the conclusions?

Reviewer #1: Yes

Reviewer #3: Yes

3. Has the statistical analysis been performed appropriately and rigorously? 

Reviewer #1: I Don't Know

Reviewer #3: Yes

4. Have the authors made all data underlying the findings in their manuscript fully available?

Reviewer #1: No

Reviewer #3: Yes

5. Is the manuscript presented in an intelligible fashion and written in standard English?

Reviewer #1: Yes

Reviewer #3: Yes

6. Review Comments to the Author

Reviewer #1: The manuscript has been substantially improved. The authors are to be congratulated on a job well done!

Reviewer #3: (No Response)

7. PLOS authors have the option to publish the peer review history of their article (what does this mean?). If published, this will include your full peer review and any attached files.

Reviewer #1: No

Reviewer #3: No

---

## [Editor Report · Acceptance letter]

22 Jun 2023

PONE-D-22-25412R2 

Effect of linezolid on platelet count in critically ill patients with thrombocytopenia 

Dear Dr. Tatsumi:

I'm pleased to inform you that your manuscript has been deemed suitable for publication in PLOS ONE. Congratulations! Your manuscript is now with our production department. 

Kind regards, 

on behalf of

Dr. Elizabeth S. Mayne 

Academic Editor

PLOS ONE